# Measuring the Diameter of Single-Wall Carbon Nanotubes Using AFM

**DOI:** 10.3390/nano13030477

**Published:** 2023-01-24

**Authors:** Dusan Vobornik, Maohui Chen, Shan Zou, Gregory P. Lopinski

**Affiliations:** Metrology Research Center, National Research Council, Ottawa, ON K1A 0R6, Canada

**Keywords:** atomic force microscopy, carbon nanotubes, diameter, nanometrology

## Abstract

In this work, we identify two issues that can significantly affect the accuracy of AFM measurements of the diameter of single-wall carbon nanotubes (SWCNTs) and propose a protocol that reduces errors associated with these issues. Measurements of the nanotube height under different applied forces demonstrate that even moderate forces significantly compress several different types of SWCNTs, leading to errors in measured diameters that must be minimized and/or corrected. Substrate and nanotube roughness also make major contributions to the uncertainty associated with the extraction of diameters from measured images. An analysis method has been developed that reduces the uncertainties associated with this extraction to <0.1 nm. This method is then applied to measure the diameter distribution of individual highly semiconducting enriched nanotubes in networks prepared from polyfluorene/SWCNT dispersions. Good agreement is obtained between diameter distributions for the same sample measured with two different commercial AFM instruments, indicating the reproducibility of the method. The reduced uncertainty in diameter measurements based on this method facilitates: (1) determination of the thickness of the polymer layer wrapping the nanotubes and (2) measurement of nanotube compression at tube–tube junctions within the network.

## 1. Introduction

Advances in methods for the scalable manufacture, purification, and dispersion of single-wall carbon nanotubes (SWCNTs) have dramatically accelerated the development of applications based on this material. In particular, various approaches for separating semiconducting and metallic tubes have enabled demonstration of a variety of SWCNT-based electronic devices [1,2,3,4,5,6,7]. The increased availability of highly purified SWCNT samples containing a limited number of chiralities has highlighted the need for improved methods for assessing these materials. With fewer chiralities present, these samples are expected to have narrower diameter distributions, motivating the need for accurate measurements of this key dimensional parameter. While transmission electron microscopy (TEM) is a powerful method to measure the diameter of individual tubes [8,9,10], this approach is costly and requires special substrates. Raman spectroscopy is commonly used to determine SWCNT diameters through the frequency of the radial breathing mode, which is related to diameter [11,12]. However, recent studies comparing diameter distributions measured by Raman and TEM methods reveal discrepancies, highlighting the limitations of the Raman approach [13,14].

Atomic force microscopy (AFM) is a widely used technique capable of measuring the vertical dimensions of nano-objects with subnanometer accuracy [15]. While AFM has been commonly used to image SWCNTs [16,17,18], fewer studies have focused on using this method for measuring the tube diameter [19,20,21]. In contrast to TEM, AFM can be performed on a wide variety of substrates, including those relevant for device applications. One limitation is that AFM-measured heights often differ from their actual height due to interactions of the AFM tip with the sample. These issues have been studied in detail for soft biological samples such as oligonucleotides, where it was shown that the AFM measurements with common settings can underestimate the actual height by over 50% [22,23,24]. In the case of SWCNTs, the effects of both substrate–tube and tip–tube interactions during AFM imaging were examined with molecular dynamics calculations, concluding that for tube diameters less than 2 nm, applied forces are not sufficient to significantly alter the measured height [19]. In this work, we show that even at the moderate forces commonly used for AFM imaging, compression of the SWCNTs does occur, which can result in a significant underestimation of the diameter extracted from the images. A straightforward experimental procedure has been developed to minimize and/or correct for this effect and extract accurate values of the SWCNT diameter. Another issue that contributes to the uncertainty of diameter measurements, the roughness of both the nanotubes and the substrate, has also been considered. A simple analysis protocol has been developed to reduce these uncertainties to <0.1 nm.

To demonstrate the utility of our protocol for AFM-based diameter measurements, we applied it to measure the diameter distribution of individual nanotubes in networks of highly semiconducting enriched polyfluorene/SWCNT dispersions on silicon oxide substrates. Good agreement was obtained for diameter distributions measured on the same sample with two different commercial AFM instruments and analyzed by two different analysts. The reduced uncertainty associated with diameter measurements using this method facilitates determination of the thickness of the polymer layer wrapping the nanotubes, which has not been reported previously. In addition, analysis of height measurements at tube–tube crossings within the network demonstrate compression at the junctions. Quantifying this compression is expected to contribute to further understanding of electron transport at these junctions that limit the performance of devices based on SWCNT networks [25]. The method for extraction of vertical dimensions of nano-objects from AFM images outlined here is not specific to carbon nanotubes but can also be applied to a variety of 1D and 2D nanomaterials to ensure that AFM measurements result in accurate diameter and/or thickness measurements.

## 2. Materials and Methods

### 2.1. Sample Preparation

Four different SWCNT dispersions were utilized to prepare nanotube networks in this study. Ultrahigh purity SWCNTs (>99.9% semiconducting) poly(9,9-di-n-dodecylfluorene) (PFDD)-wrapped nanotubes (IsoSol S-100) were purchased from NanoIntegris (Boisbriand, QC, Canada) in two forms, a PFDD/SWCNT dispersion (10 mg/L) in toluene and solvent-free bucky paper which was redispersed in toluene to the same nanotube concentration. The diameter range of the SWCNTs in both these samples are specified as 1.2–1.4 nm. The polymer-to-nanotube ratio was measured by UV–Vis–IR absorption to be 4:1 and 1:1, respectively. Semiconducting SWNCTs in an aqueous surfactant solution (IsoNanotubes-S 99%, diameters of 1.2–1.7 nm) were also obtained from NanoIntegris. The final tube type measured was SWCNTs dispersed in dimethyl sulfoxide (DMSO) at 100 mg/L (SEER ink, Linde North America, Bridgewater, NJ, USA), for which the diameter range was given as 1.4–2 nm.

The PFDD/SWCNTs in toluene and aqueous SWCNT dispersions were deposited onto piranha-cleaned ~1 cm^2^ silicon substrates (Si(100) with 100 nm thermal oxide, Silicon Quest International, San Jose, CA, USA). PFDD/SWCNT dispersions were deposited directly onto clean SiO_2_ for 5 to 15 min, followed by rinsing with toluene for 20 to 60 s and drying with nitrogen. Aqueous surfactant SWCNT dispersions were deposited on Poly-L-Lysine (PLL)-coated SiO_2_ for 15 min, rinsed with water, and dried with nitrogen. The Linde SWCNT dispersion in DMSO was handled in a nitrogen-purged glovebox, where 100 µL of 2 mg/L DMSO-nanotube dispersion was deposited on a freshly cleaved 1 cm^2^ square of highly oriented pyrolytic graphite (HOPG). The sample was then heated to 100 °C for approximately 15 min, causing the DMSO to evaporate. For all samples, AFM measurements were performed within a day of preparation, but several samples were measured multiple times at different dates during several months following their preparation with no significant changes in overall morphology or measured diameters.

### 2.2. AFM Measurements

Most of the AFM data were obtained with a MultiMode AFM with a NanoScope V controller (Bruker Nano Surfaces Division, Santa Barbara, CA, USA) in Peak Force QNM mode. In this mode, the AFM tip oscillates at a given (2 kHz) frequency, with the highest (peak) repulsive interaction force used as the feedback signal. Silicon nitride ScanAsyst-Air AFM probes (Bruker AFM Probes, Camarillo, CA, USA) were used in all peak force feedback measurements. The AFM tip diameter and cantilever spring constants (as specified by the supplier) were 2 nm and 0.4 N/m, respectively.

To demonstrate the wide applicability of the measurement methods presented here, a Nanowizard II BioAFM (JPK Instruments, Berlin, Germany) in intermittent contact mode was also used to measure SWCNT diameters. Silicon cantilevers (HQ:XSC11/AL BS, Cantilever D, typical radius of 8 nm, resonance frequency of 350 kHz, spring constant of 42 N/m, MikroMasch, Watsonville, CA, USA) were used for imaging in air at RT (20–23 °C). Force was carefully minimized by first gradually reducing the free amplitude to the highest level where stable imaging could be performed (evaluated by comparing trace–retrace lines while scanning the sample) and adjusting the feedback set point to a value as close as possible to the free amplitude. The typical set point obtained in this way was 85–90% of the free amplitude, which was set to 600 mV.

Images used to measure diameters were typically 1 × 1 µm^2^ in size, acquired with 512 × 512-pixel resolution and at a scan rate of 1 Hz. In this way, the lateral pixel size was approximately 2 × 2 nm^2^. The measured diameters of single nanotubes were in the range from 1.2 to 2 nm, and based on our tests, the radii of AFM tips for peak-force AFM were in the 2 to 15 nm range (tips were replaced when their radii exceeded 15 nm, even if they were still scanning properly). Based on the combination of the tip and nanotube radii, the pixel size of 2 × 2 nm^2^ was deemed sufficient to have reliable nanotube height measurements. The pixel size should not be larger than the smallest tip size since that could lead to two adjacent pixels across the nanotube being both off the highest point of the nanotube, leading to underestimation of the tube height.

Traceable calibration grids (STS3-180P, STS3-440P, STS3-1000P, and STS3-1800, VLSI Standards Inc., Milpitas, CA, USA) were used on a regular basis (approximately every 6 months) for calibration verification and adjustments of our AFM instruments. On top of that, we regularly imaged atomic steps on HOPG. These steps have a known height of 0.34 nm. In our tests, we always measured the height of the smallest steps to be 0.34 nm +/− 0.01 nm, consistent with the expected value, further confirming the precision of our calibration (see Appendix A). The step height is much closer to the value of the SWCNT diameters than that of the calibration grids, thus ensuring that the AFM performs accurately at the relevant height scales.

### 2.3. Analysis of AFM Data

All image processing and analyses were performed using Gwyddion (v2.45, Czech Metrology Institute), a free, widely available open-source software [26]. For the intermittent contact mode AFM, images were additionally flattened using the first-level flattening with the JPK Data Processing Software (v5.1.8, JPK instruments, Berlin, Germany) prior to using Gwyddion software for further image processing. Image processing consisted of flattening the background to correct for drifts and zeroing the z scale. Images were flattened using the polynomial background removal function in Gwyddion, where special care was taken to only use the pixels corresponding to the substrate for the flattening and not those corresponding to the nanotubes or contaminants. This was ensured by first setting a mask based on a height threshold highlighting any features higher than the substrate, and then specifying to use only the unmasked pixels for the polynomial fitting and flattening.

## 3. Results and Discussion

### 3.1. Applied Force in AFM Imaging Leads to Nanotube Compression

To investigate the effect of applied force on measurements of nanotube diameter, the force was gradually increased while imaging the same network area. Figure 1 illustrates one such test on a network prepared from a 4:1 PFDD/SWCNT dispersion deposited on a SiO_2_ substrate. The lowest force that consistently resulted in stable imaging, regardless of the AFM tip or of the particular nanotube sample, was 0.2 nN. Using this as the initial force, sequential images were obtained at increasing peak-force feedback values followed by a final image back at 0.2 nN, as shown in Figure 1a. It is apparent from the figure that even at the highest AFM applied force used here (2 nN), the imaging remained non-destructive as the nanotubes remain intact and do not appear to have been moved during imaging. Upon reducing the peak force back to 0.2 nN, the image appears essentially identical to the initial one, indicating that the tip was kept unchanged while acquiring the entire set of images. The maximum of the attractive van der Waals force between the tip and the sample was also monitored, remaining at −50 pN. Since the attractive van der Waals force is directly proportional to the tip diameter, this indicates that the tip was unchanged [27].

For each image in Figure 1a, the average network height was determined as described previously [27]. This was done rapidly by averaging the height of all the pixels above a certain height threshold as indicated by the colored mask (Figure 1b), which identifies the pixels corresponding to nanotubes. Once the network–substrate separation height threshold is selected, the Gwyddion software allows a one-click extraction of the average height of all colored pixels (nanotube network height), as well as the average height of all non-colored pixels (substrate average height). The final average network height is obtained by subtracting the average substrate height from the network height. Figure 1c illustrates that increasing the force decreases height in a linear manner, facilitating extrapolation of the data to extract the height at zero applied force. Control experiments involving imaging terraces on HOPG show no change in step height for a similar range of applied forces (see Appendix A).

Figure 2 shows average height versus applied force data for four different commercially available SWCNT samples (representative images are shown in Appendix A). Despite the differences in nanotube types and diameters, all graphs show a force-dependent behavior. Linear fits to this data, shown in Figure 2, allow extraction of the corresponding zero-force height and the “compression” slope. The differences in compression slopes can be qualitatively interpreted as a result of the differences between the four samples (polymer or surfactant on surface of the tubes, different diameter distributions, etc.), but more data are required to draw firm conclusions. Here, we only note that independent of the details of a particular carbon nanotube sample, the forces typically applied in AFM imaging (e.g., 0.2–2 nN) can significantly compress the tubes, leading to potential errors in measured diameters.

The radial deformability of CNTs in AFM imaging has been demonstrated previously within a similar force range [28,29,30], but this compression effect has often been neglected in studies where AFM was used to measure diameters of carbon nanotubes [19,21]. Our results clearly indicate that imaging of SWCNTs results in compression of the tubes, translating to an underestimation of the nanotube diameter, even at the moderate forces commonly employed in AFM measurements. This compression effect can lead to errors easily reaching 30% or more of the actual nanotube height. However, measuring the height of the tubes as a function of applied force allows extraction of the average height of the tubes at zero force. The data shown in Figure 1 and Figure 2 also indicate that for our set-up, using a force of 0.2 nN or less minimized this compression error.

### 3.2. Substrate and Nanotube Roughness Contribute to Diameter Uncertainty

While the method used above is useful for rapidly demonstrating the average diameter of an ensemble of nanotubes, it does not permit measurement of the diameter of an individual SWCNT (except in the case where there is only a single tube in the image). The conventional method to determine the heights of individual features from an AFM topography image involves extracting a height profile perpendicular to the object being measured. Here we show how this approach can lead to significant variations in the measured diameter due to the roughness of both the substrate and the nanotube.

Figure 3 shows a typical 1 µm^2^ AFM image of a PFDD/SWCNT network on SiO_2_ along with parallel pairs of height profiles extracted from the measured data. For each pair, one profile is along the top of the nanotube and the other is on the substrate adjacent to the tube. The profiles on the bare SiO_2_ substrate (the red points in the figure) are observed to be more uneven, with height variations sometimes exceeding 1 nm. The standard deviation of the substrate height profile data ranges from 0.31 to 0.18 nm for the four profiles shown in the figure. The nanotube profiles (blue points) also appear uneven, but with considerably smaller height variations that seldom exceed 0.5 nm and with standard deviations ranging from 0.13 to 0.10 nm. While this is less roughness than that of the substrate, it is still significant and appears to be uncorrelated to that of the nearby substrate. This is consistent with studies quantifying the persistence length of carbon nanotubes, showing they behave like rigid rods within a length range that exceeds one hundred nanometers [31] and thus do not conform to shorter range fluctuations in the substrate height shown here. The height variations on the nanotubes are also not periodic, as would be expected if the polymer was wrapping the tube in a helical manner [32]. The height variations on the tubes exceed the random noise limits of our AFM setup, as height measurements with standard deviations below 0.1 nm are typically observed on atomically flat surfaces such as HOPG or mica at the same imaging parameters. Moreover, we observed similar tube roughness for all four different SWCNT samples studied. While the source of this roughness cannot be definitively assigned, for PFDD-wrapped nanotubes, it probably arises from variations in the conformation of the polymer coating the tube. In the case of the IsoNanotubes-S or the Linde SEER tubes, it is likely due to the surfactant or salt, respectively, used to disperse the tubes in the solution. Atmospheric contaminants binding to the tubes may also contribute to the observed roughness.

The profiles in Figure 3b–e show that the combined roughness of the substrate and nanotubes can lead to significant inaccuracies if the conventional AFM diameter measurement method is used. This method involves extracting cross-sections perpendicular to the nanotubes and measuring the height of the substrate from that of the highest point on the nanotube. Examining the profiles in Figure 3 shows that the difference between the tube apex and the substrate can vary substantially at different points along the tube, with some examples of these differences indicated in the figure. Although the difference between the average heights of the tube and substrate for the four profiles shown range from 1.7 to 2.2 nm, the individual height differences measured at each x-value range from 1 to 2.5 nm. This illustrates how large errors could result from an overly simplified approach to diameter measurements.

### 3.3. Analysis Method to Rapidly Extract SWCNT Diameters

One way to decrease the diameter measurement errors that result from roughness is to use an averaged diameter value based on multiple cross-sections. This can be accomplished easily using the Gwyddion analysis software, which offers an automated way to average up to 128 contiguous, single-pixel-wide cross-sections in one click. We find that using even 10-pixel-wide cross-sections reduces the uncertainty significantly, with larger widths offering even greater reduction. To further minimize this uncertainty, the averaging of the substrate baseline height can be improved by utilizing all the substrate pixels over the entire image. This can be done by choosing a height threshold to select the nanotube features (define a mask), as described above. Then, the average height value of all the pixels that are not masked is used as the average height of the substrate, which is then subtracted from the maximum height of the averaged nanotube profile to obtain the diameter.

The decrease in uncertainty realized by using this averaging analysis method was evaluated by randomly selecting an AFM image of a SWCNT network on SiO_2_, and then repeating the analysis on it several times, each time extracting the diameters of the same tubes. In each of the images, diameters of a number of nanotubes (a total of 20) were measured 5–7 times using the above-described methods. Finally, the diameter standard deviation was calculated for each of the nanotubes using all of the extracted diameter values for that nanotube. The same diameter values for each of the nanotubes could be reliably obtained, with the standard deviation reduced to below 0.1 nm (see Appendix A for an example of this evaluation). This analysis procedure is effective in that it allows for a greater degree of confidence in the measured diameter values with minimal additional analysis.

### 3.4. Diameter Measurements of PFDD/SWCNTs

Upon designing an effective method to increase the accuracy of the AFM measurements of SWCNT diameters by optimizing both our experimental (minimized forces) and analysis (averaging) approach, we set out to apply it to measure the diameter of PFDD/SWCNTs, providing insight into the interaction of the polyfluorene polymer with the nanotubes. PFDD/SWCNT networks were formed on SiO_2_ substrates as described in the experimental section, and AFM images were obtained using a minimized peak-force value of 0.2 nN. Averaged cross-sections of 120 single nanotubes were then extracted using 30-pixel-width cross-sections. For each nanotube cross-section, the averaged substrate level was subtracted from the maximum height to obtain the diameter. To verify the universality of this approach, the same sample was also imaged with a different AFM instrument (see the Methods section for details), using the conventional intermittent contact mode with amplitude-based feedback. For these measurements, the AFM applied force was minimized by minimizing the free amplitude, and then using the smallest amplitude set point that facilitated stable imaging. The same analysis method (averaged cross-sections and average substrate baseline extraction) was used to measure diameters of an additional 126 nanotubes. Figure 4 shows the distribution of diameters that were obtained on the two instruments (with two different analysts carrying out the analysis). The blue-colored histogram corresponds to the peak-force AFM measured diameters, and the red histogram corresponds to the intermittent contact-mode AFM measurements. The two histograms are remarkably similar, with the average diameter value of 1.75 ± 0.23 nm from the peak force data and 1.67 ± 0.22 nm obtained using the intermittent contact mode AFM. With both methods, the measured diameter values ranged from 1.1 nm up to 2.2 nm.

The nanotubes in the PFDD/SWCNT dispersion have been previously measured to have a narrow diameter distribution ranging from 1.2 to 1.4 nm [33]. Raman observations of the radial breathing mode for samples prepared in this work confirmed that the nanotube diameters were in this range (data shown in Appendix A). The larger average diameter obtained by the AFM measurements indicates that the polymer remained on the majority of the tubes even after all the preparation and rinsing procedures. While there was a range of diameters observed indicating that some (very few) nanotubes may be bare, while others carried more polymer, we can estimate the average added polymer thickness by subtracting 1.3 nm (the median of the diameter range for the unmodified nanotubes) from the average of the measured diameter values. Based on these measurements, the estimated average polymer thickness was in the range of 0.45–0.37 nm, suggesting a continuous coating of polymer along the nanotubes.

### 3.5. Measuring Compression at SWCNT Junctions

The protocol for accurate measurements of carbon nanotube heights can also be used to investigate junctions between tubes in these random networks. These junctions are of interest since electron transport across these junctions is the factor limiting the conductivity of random nanotube networks. Compression at SWCNT junctions has been modelled theoretically [34] and observed experimentally [35] in early studies on unmodified tubes.

To measure compression at a nanotube junction, the maximum height at the crossing point is compared with the sum of the two diameters of the tubes that cross. Figure 5 shows an AFM image depicting several junctions. Cross-sectional height profiles extracted from this image to measure the height of the junction (black) are also shown, along with the height of the two individual tubes (red and blue) making up the junction. The height of the junction is seen to be considerably smaller than the sum of the individual tubes. A histogram showing the comparison of measured and expected heights for 20 such junctions is shown in Figure 5c. All the junctions, apart from one case (junction #2 in the histogram), show a measured junction height that is smaller than expected, with the height reduction ranging from 8−34%. Averaging over all twenty junctions, the amount of compression is determined to be 19 ± 8%, similar to the previous observation [35]. Of the junctions analyzed, most are between single tubes based on the expected height being less than ~4 nm, although some with larger heights (such as junctions #3, #9, #10, and #17) likely involve small tube bundles. In the case of junction #2, which showed no significant compression, the tubes are seen to cross at a small angle. Additional data are required to determine the dependence of compression on the crossing angle.

## 4. Conclusions

Two issues that can significantly affect the accuracy of AFM height measurements of single-wall carbon nanotubes have been identified. Measurements on several different SWCNT samples indicate that under forces commonly used for AFM imaging, nanotubes can be readily compressed, leading to a significant underestimation of the diameter. A simple procedure, consisting of imaging the same area several times at different forces and plotting height vs. force graphs, can verify the extent of the compression effect and recover the actual nanotube height via extrapolation to zero-force height. In addition, we showed that the common AFM height analysis method based on drawing a cross-section across each of the nanotubes to extract the diameter of nanotubes can lead to significant errors arising from the roughness of both the nanotubes and the substrate. An analysis protocol based on averaging a larger number of pixels was proposed and shown to be easily implemented using a free and open-source SPM analysis software. This enables decreasing the variation of measured heights to less than 0.1 nm. It is important to note that these results were obtained for moderately dense semiconducting SWCNT networks on SiO_2_ that are functional as a channel material in transistors. While lower uncertainties may be achieved for isolated, pristine (no polymer or surfactant used for purification and dispersion) nanotubes deposited on ideally flat substrates such as mica or HOPG, the goal here was to develop a method for diameter measurements on “real-world” samples such as those used for device fabrication.

To illustrate the utility of this diameter measurement method, it was applied to measure the diameter of commercially available, highly enriched semiconducting nanotubes. The measured diameter for the nanotubes in a PFDD/SWCNT sample was shown to be larger than expected based on the diameters of the tubes present in the dispersion, suggesting the polymer remained on the SWCNTs upon deposition on the substrate and rinsing with a solvent. The average thickness of the polymer on the nanotubes was estimated to be ~0.4 nm. In addition, measured heights of tube–tube junctions were found to be less than the diameter of the two tubes that are crossing, indicating compression of ~20% at the junctions. Finally, we note that the measurement protocol developed here should also be generally applicable to measurement of vertical dimensions of other 1D or 2D nanomaterials.

## Figures and Tables

**Figure 1 nanomaterials-13-00477-f001:**
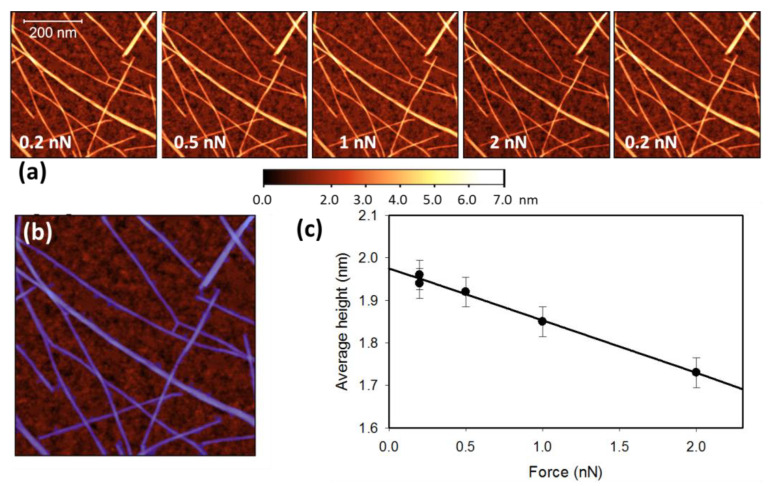
(**a**) Consecutive images of the same area of a PFDD/SWCNT network on a SiO_2_ substrate, obtained at different peak-force feedback values. (**b**) The blue mask shows all the pixels whose height was included in obtaining the average network height. (**c**) Average network height as a function of applied force along with a linear fit to the data.

**Figure 2 nanomaterials-13-00477-f002:**
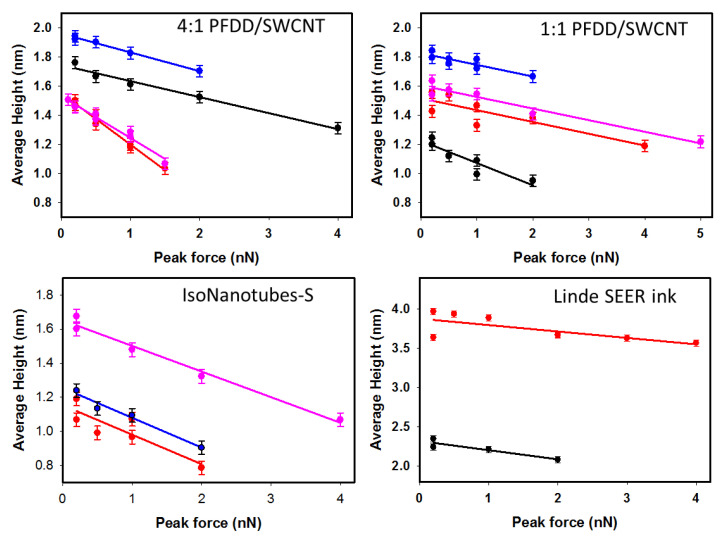
Average carbon nanotube network height as a function of AFM peak force measured on several different areas of four different SWCNT samples. In each graph, data points of the same color are obtained from consecutive AFM images of the same area.

**Figure 3 nanomaterials-13-00477-f003:**
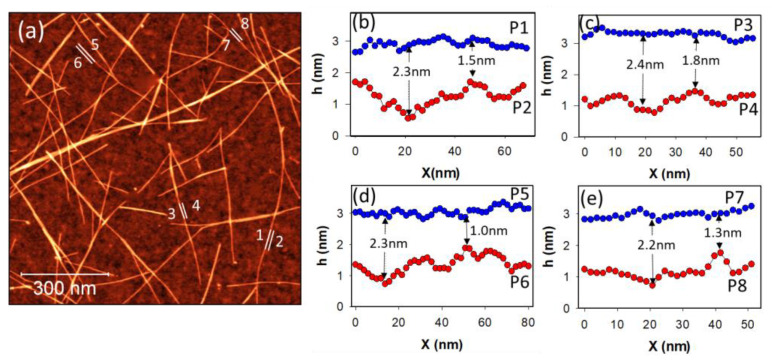
Roughness of profiles on nanotubes and on the SiO_2_ substrate. (**a**) A typical 1 × 1 µm^2^ AFM image of a network made of PFDD/SWCNTs with numbered white lines, along which the profiles P1–P8 shown in (**b**–**e**) were extracted. The data points corresponding to profiles along the top of the nanotubes are shown in blue, while adjacent substrate profiles are shown in red. Differences in the height of the substrate and the nanotube at selected points are indicated to illustrate the variability.

**Figure 4 nanomaterials-13-00477-f004:**
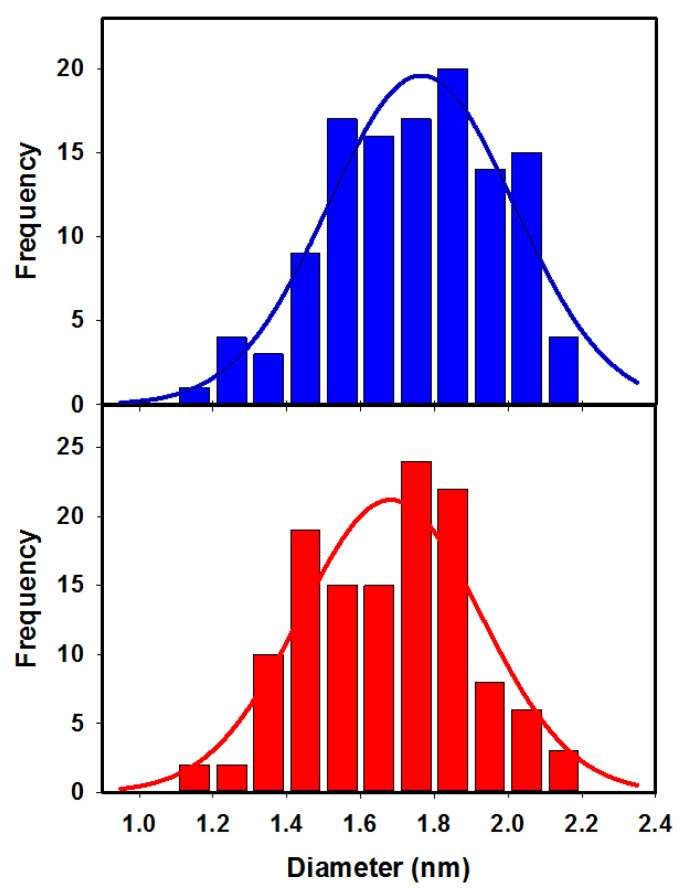
Histograms of the distribution of diameters measured on PFDD/SWCNTs using peak-force (Bruker) AFM (**blue**) on 120 nanotubes and intermittent contact-mode (JPK) AFM (**red**) on an additional 126 nanotubes. The red and blue solid lines represent Gaussian fits to the respective histogram data.

**Figure 5 nanomaterials-13-00477-f005:**
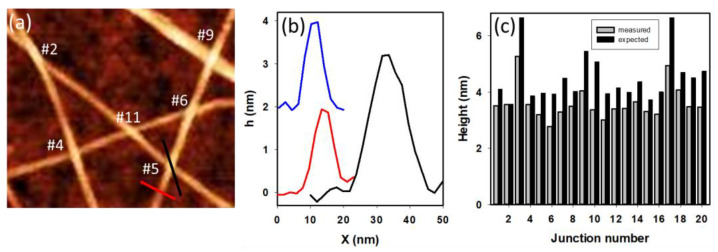
(**a**) AFM image of PFDD/SWCNTs showing several nanotube crossings. (**b**) Height profiles across the individual tubes (blue and red) and the crossing point (black) for junction #5. (**c**) Histogram showing comparison of measured junction heights and expected values based on the sum of the individual tube heights.

## Data Availability

Data available upon request.

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
