# Peer review of "Measuring the Diameter of Single-Wall Carbon Nanotubes Using AFM"

_nanomaterials, 2023, doi:10.3390/nano13030477_

Round 1

Reviewer 1 Report

1. L.180: Figure 1(d) does not exist. It is probably (c). 

2. Figure 2 shows the average data. It would be more helpful to have error bars.

3. Do you think the chirality of SWNTs affects the relationship between the force and the height?

4. There are several typos. Please check carefully and correct them.

Author Response

We thank the referee for their comments on our manuscript. 

1) We have corrected the error. It is indeed Fig. 1(c)

2) Error bars have been added to Fig. 2. 

3) This is an interesting question which would require further study beyond the scope of this paper. In particular, single chirality samples would likely be required for a definitive study of how the compression effect depends on chirality. 

4)  We have corrected a number of typos found in the manuscript. 

Reviewer 2 Report

This is a very well written and interesting paper, and the precise methodology described by the authors will help others in the field to use AFM for a much more accurate measurement of SWCNT diameters. Prior to acceptance, I only have one request of the authors - I would like to see a complementary technique like Raman spectroscopy be used on a small and dilute sample of SWCNTs on a substrate. I'd like to see how the diameter-dependent Raman radial breathing mode frequencies match the diameters determined by the AFM study.

Author Response

We thank the referee for comments on our manuscript. The suggestion to use a complementary technique such as Raman is a good one. We have obtained Raman measurements of the radial breathing mode on a PFDD/SWCNT sample used in the current work. The frequency of the RBMs are consistent with the 1.2-1.4 nm diameter range measured previously for these nanotubes. A comment has been added to the manuscript on p. 8 and the data has been added in the supporting information (Fig. S4).  

Reviewer 3 Report

It is rare to find such a well written paper. I have only found one mistake to be corrected: on line 180 it is written Figure 1(d), I could not find (d) in Figure 1. Please do correct!

Author Response

We thank the referee for comments on our manuscript. The error was also noted by Reviewer 1 and has been corrected. 

Reviewer 4 Report

This paper told us that the measurement of single-walled carbon nanotube diameters using atomic force microscopy.

Taken into account all the following shortcomings mentioned, this paper needs to be revised.

1. The abstract section is suggested to be rewritten, and data can be added to verify the accuracy and reliability of single-walled carbon nanotube diameter measurements using AFM, add linking words to enhance readability, and most importantly, highlight innovation and application.

2. It is suggested that the first part of the general description of the paper should highlight the innovation and importance of the paper.

3. The generality and universality of the method for measuring the diameter of single-walled carbon nanotubes described in this article should be clearly described.

4. The 2.1 Sample preparation part of the content should be more streamlined.

5. The description of the experimental phenomena should be objective facts, while reducing speculative terms, and it is recommended that the whole article be revised in scope.

6. The layout of the three pictures in Figure 1 is not beautiful and should be adjusted.

7. The first two figures in Figure 1 clarity needs to be improved, all the pictures in the text can be adjusted to be larger in order to facilitate observation, the current feeling is too small to affect the data reading.

8. The third part of the analysis of the causes of the experimental phenomenon and the trend of the picture is not detailed enough, it is recommended to analyze the data in a deep anatomy.

9. The analysis for Figure 2 is too little, and the final results obtained should be highlighted.

10. For Figure 4 and Figure 5, the color matching and picture details such as the thickness of the lines are not in place, it is recommended to adjust to make the reader get a better sense.

Author Response

We thank the Reviewer for the comments on our manuscript. We have considered all the points raised and have made substantial revision to the manuscript. In particular we have revised all the figures to improve the readability of the data. In addition we have been significant changes to the text throughout the manuscript to make it easier to follow. We believe the manuscript has been improved as a result. With respect to the specific points noted by the referee we have responded as follows.  

1) The abstract was re-written in order to more clearly summarize the contents of the manuscript. 

2) We have considered the suggestion to revise the introduction but have kept the introduction in its current form. Currently the introduction has three paragraphs; 1) Explains motivation for improved measurements of nanotube diameter, 2) Discusses previous work using AFM for dimensional measurements of nano-objects, placing the work in the context of previous studies and 3)  Summarizes the main results of the current work and its implications and significance. 

3) The generality of the current method is demonstrated in section 3.4 where we show good agreement for diameter measurements obtained on the same sample with two different AFM instruments. We have added this point to abstract to emphasize the importance of this result. 

4) Section 2.1 has been edited extensively to streamline and enhance readability. We note that sample preparation in this work is complicated as four different types of SWCNT samples were used and we need to provide sufficient information on this to ensure our results can be reproduced. 

5) We have gone through the manuscript to ensure that facts and speculations are clearly separated. In addition to the facts supported by the observations it is reasonable to offer interpretations of these observable facts. 

6+7) Figure 1 has been modified. All the figures have been modified and enlarged where possible to make them clearer and more readable. 

8)  Not sure what section of the manuscript is being referred to here. We have revised Fig. 3 and the associated discussion of this Figure to make it easier to follow. 

9) The main result of Fig. 2 is to show that AFM compresses tubes of several different types under typical imaging conditions. This is clearly stated in the manuscript. 

10) Fig. 4 and 5 have been revised to make them easier to follow. 

Round 2

Reviewer 2 Report

The authors have addressed our comments and this manuscript is ready for acceptance.

Reviewer 4 Report

This paper told us that the measurement of single-walled carbon nanotube diameters using atomic force microscopy.

According to the suggestion before, the authors have revised this article including the language such as grammar and format, interpunction, the authors also gave the reasonable expression about the questions.

The manuscript can be accepted.
